# Psychosocial Factors Associated with Self-Management in Patients with Diabetes

**DOI:** 10.3390/healthcare11091284

**Published:** 2023-04-30

**Authors:** Rodrigo León-Hernández, Andrea C. Rodríguez-Pérez, Yessica M. Pérez-González, María I. P. de Córdova, Raúl de León-Escobedo, Tranquilina Gómez-Gutiérrez, Filiberto Toledano-Toledano

**Affiliations:** 1Consejo Nacional de Ciencia y Tecnología, Av. de los Insurgentes Sur 1582, Crédito Constructor, Benito Juárez, Mexico City 03940, Mexico; 2Facultad de Enfermería Tampico, Universidad Autónoma de Tamaulipas, Adolfo López Mateos S/N, Universidad, Tampico, Tamaulipas 89109, Mexico; 3Facultad de Medicina de Tampico, “Dr. Alberto Romo Caballero” Universidad Autónoma de Tamaulipas, Tamaulipas 89000, Mexico; 4Unidad de Investigación en Medicina Basada en Evidencias, Hospital Infantil de México Federico Gómez Instituto Nacional de Salud, Dr. Márquez 162, Doctores, Cuauhtémoc, Mexico City 06720, Mexico; 5Unidad de Investigación Multidisciplinaria en Salud, Instituto Nacional de Rehabilitación Luis Guillermo Ibarra Ibarra, Calzada México-Xochimilco 289, Arenal de Guadalupe, Tlalpan, Mexico City 14389, Mexico; 6Dirección de Investigación y Diseminación del Conocimiento, Instituto Nacional de Ciencias e Innovación para la Formación de Comunidad Científica, INDEHUS, Periférico Sur 4860, Arenal de Guadalupe, Tlalpan, Mexico City 14389, Mexico

**Keywords:** diabetes, self-management, self-efficacy, social support, depression, structural equation modeling, SEM

## Abstract

Despite the significant advances in research on diabetes, relatively few researchers have examined the theoretical and empirical usefulness of explanatory models that contribute to self-management of the disease. In response to the theoretical and empirical approaches related to this topic, the objective of this research was to assess a hypothetical model to explain self-management behavior in patients with type II diabetes through structural equation modeling in a population of users of the services of the State Health Department of Tamaulipas, Mexico. The study used a cross-sectional and explanatory design. The sample was intentional. A total of 183 patients with a diabetes diagnosis completed a sociodemographic data questionnaire, the Partners in Health Scale, the Duke-UNC-11, the Family Apgar, the Self-Efficacy Scale, the Personal Health Questionnaire and the Physical Activity Scale. The results indicated that the hypothetical model was improved by excluding the exercise variable. The appropriate model was used to determine the effects of depression, social support, self-efficacy, family functioning, years of formal education and years with a diagnosis on self-management. The goodness-of-fit indices (GFIs) were good, i.e., χ^2^/gl = 0.89 (*p* = 0.529), root mean square error of approximation (RMSEA) = 0.000, and comparative fit index (CFI) = 1.000, with an acceptable degree of parsimony (PNFI = 0.409 and PGFI = 317). The model explained 33.6% of the variance. Therefore, this model represents an important advance in knowledge concerning self-management and provides empirical and theoretical evidence, particularly for the Mexican or Latino population.

## 1. Introduction

It is estimated that in 2021, 537 million people lived with diabetes, and an economic cost of USD 966 million was generated [1]. The National Health and Nutrition Survey (2018–2019) revealed that 14.4% of older adults in Mexico have diabetes, and this percentage exceeds 30% in the population aged over 50 [2].

Faced with the problems that this disease presents, the Pan American Health Organization promotes the Chronic Care Model in the Americas that comprises six components, including support for self-management, which is conceptualized as patient education and health interventions by personnel that can strengthen patients’ skills in terms of the proper management of this chronic disease [3,4].

To understand how self-management works, it is important to consider both elements of the health professional–patient dyad. One element of the dyad is the health professional who provides self-management support, and the other element is the patients who conduct their own self-management behavior. The patient is the main focus of this study. Self-management behavior involves people playing a central role in the management of their disease and participating in the treatment of and education regarding their health condition. The treatment and education include biological, emotional, psychological and social aspects [5]. In addition, it is important to consider the individual and family self-management theory, which proposes that self-management is a dynamic phenomenon made up of three dimensions: context, process and results. The context influences individual and family participation in the development of self-management and impacts positively on the results, whether proximal or distal. However, individual and family processes occur in this context and can act as risk or protective factors for the development of positive self-management [6].

Empirical research has used regression analyses to identify the following variables that can predict self-management: belief in the effectiveness of treatment, expectation of results, family support and self-efficacy [7], the doctor-patient relationship and social support [8], sex (female), age and body mass index [9], years with the disease, education level and diabetes education [10], knowledge of the disease, patient activation and depression [11,12,13].

Similarly, various models have been developed through path analyses to explain the interaction between self-management behavior and other variables. The predictors include age, disease duration, knowledge and attitudes toward the disease [14]. Preceded by self-efficacy and health literacy, self-management has indirect effects on health-related quality of life [15]. Moreover, the direct influence of self-management on glycemic control and the perception of health status has been documented [16,17].

As shown in the literature, the interactions among the variables that influence self-management behavior are complex. Therefore, it vital to conduct studies that contribute to improving understanding of the development of self-management, particularly in the Latino population, among whom the subject has rarely been studied. In Mexico, the variables that predict self-management behavior have been identified through linear regression analyses in different samples of people with chronic diseases [18,19].

Therefore, the objective of this research was to assess a hypothetical model to explain self-management behavior in patients with type II diabetes through a structural equation model (SEM) in a user population of the State Health Department of Tamaulipas, Mexico.

A hypothetical path model was designed based mainly on evidence from studies conducted by members of the Chronic Diseases Self-Management Thematic Network (Red Temática de Automanejo en Enfermedades Crónicas) research team. In these studies, the following variables were identified through linear regression analyses as independent predictors of self-management behavior in patients with noncommunicable diseases (e.g., diabetes, arterial hypertension, obesity): depression, social support, self-efficacy, family functionality, exercise, years of formal education and years with a diagnosis [18,19]. Similarly, the design of the hypothetical model was supported by empirical evidence from the Chronic Disease Self-Management Program, which has shown benefits in relation to health status, healthy behaviors, and a decrease in both hospitalizations and days spent in hospital in a heterogeneous population of people with different chronic diseases, such as heart disease, lung disease, stroke or arthritis [20]. Likewise, it was based on the theory of self-efficacy developed by Bandura [21], which in its further development proposes that efficacy expectations derive from four sources: vicarious experience, verbal persuasion, physiological states and performance achievements. These factors influence cognitive efficacy processing through enactive, vicarious, hortatory, and emotive sources. The empirical evidence related to said theory supports the relationship between perceived self-efficacy and behavior changes such as self-management behavior. In this case, the latter refers to self-management behavior. The next logical step was to create a hypothetical path model to obtain empirical and theoretical evidence to advance knowledge regarding self-management behavior (see Figure 1).

## 2. Materials and Methods

### 2.1. Participants

This nonprobabilistic sample derived via intentional sampling included 183 patients with a diagnosis of type II diabetes who were users of health centers in Ciudad Victoria (n = 88) and Tampico (n = 95) in the state of Tamaulipas, Mexico. This sample included patients who agreed to participate in the study in the period from November 2018 to March 2019. The mean age of the women was 55.03 years, and the mean age of the men was 66.80 years. The inclusion criteria for this study were a medical diagnosis of type II diabetes for at least 3 months and having reached the age of maturity established in Mexico (18 years old). The patients were invited by their doctors and nurses to participate in the study. Those who accepted the invitation were referred to the principal investigators of this study. People who could not read or write and therefore could not answer the questionnaires were excluded from the study. The study design was cross-sectional and explanatory.

### 2.2. Measuring Instruments

A questionnaire was used to collect sociodemographic data, including the participants’ age, sex, marital status, education and years since diagnosis [22]. Notably, in all the measurement instruments, the indications for qualification and interpretation of the original authors and those indicated in the validation in the Mexican population were respected.

Self-management was measured with the Partners in Health Scale (PHS) developed by Battersby et al. [23]. The version validated in the Mexican population comprises 12 items divided into 3 dimensions, i.e., knowledge (2 items: 1 and 2), adherence (6 items: 3–8) and symptom management (4 items: 9–12), with a numerical response format ranging from 0 to 8 in progressive order. The categorization is based on the total scores as follows: low self-management ≤73 points, medium self-management 74 to 88 points, and self-management ≥89 points. The internal consistency was reported to be good (α = 0.88) [24,25].

Social support was assessed with the Duke-UNC-11 developed by Broadhead et al. [26]. This instrument comprises 11 items with a Likert-type response scale ranging from 1 to 5 (1 = much less than I wish/want to 5 = as much as I wish/want). This tool was validated in the Mexican population, with a single factor explaining 58.6% of the variance and good internal consistency (α = 0.92). The categorization was based on the total score as follows: absence of support ≤32 and presence of social support ≥33 [27].

Family functioning was measured with the Family Apgar developed by Smilkstein [28]. This tool contains 5 items with a Likert-type response scale ranging from 0 to 4 (0 = never to 4 = always), with the following categorization based on the total score: normal family functionality, 17 to 20 points; mild dysfunction, 16 to 13 points; moderate dysfunction, 10 to 12 points; and severe dysfunction, ≤9 points. Its validation in Spanish achieved good internal consistency (α = 0.80) [29].

Self-efficacy in the self-management of chronic diseases was assessed with the Self-Efficacy Scale developed by Lorig et al. [30]. This tool comprises 6 items with a numerical visual response scale ranging from 1 to 10. The average score is used, and “the higher the score, the greater the self-efficacy”. In Mexico, good internal consistency was obtained (α = 0.93) [25].

Depression was measured with the Personal Health Questionnaire (PHQ) developed by Ory et al. [31]. The version validated in Mexico comprises 8 items with a Likert-type response scale with 4 options (0 points = never, 3= almost every day). The categorization was based on the overall score as follows: no depressive symptoms, 0–4 points; mild depressive symptoms, 5–9 points; moderate depressive symptoms, 10–14 points; severe depressive symptoms, 15–19 points; and serious depressive symptoms, 20–24 points. In Mexico, the internal consistency was α = 0.781 [25].

Exercise was assessed with the Physical Activity Scale developed by Lorig et al. [30]. This instrument comprises 6 items that assess exercise frequency in minutes during a week, with 5 response options ranging from 0 = none to 4 = more than 3 h. The average score is used for interpretation. In its adaptation to the Mexican context, the internal consistency was α = 0.62 and the test–retest reliability was 0.72 [25].

### 2.3. Procedures

The authors of this study administered the instruments to patients with diabetes. The surveys were conducted in the waiting rooms of the health centers of the Tamaulipas Health Department from November 2018 to March 2019. The completion time was approximately 35 min. All the patients were invited to participate in the study on a voluntary basis. The research objective was read to the patients, and concerns were addressed and resolved. Those who agreed to participate signed an informed consent form and responded to the instruments in a single session. To avoid missing values, the interviewers verified that the instruments were answered in their entirety.

### 2.4. Ethical Considerations

This study is part of a research project titled “Red Temática de Automanejo en Enfermedades Crónicas” (“Chronic Diseases Self-Management Thematic Network”), CONACYT No. 293635, which was approved on 19 June 2018 by the Health Department of the state of Tamaulipas (registration number: 113/2018/SCEI) and on 11 December 2019 (registration number: 004-2019) by the Ethics and Research Committee of the School of Nursing of Tampico, Autonomous University of Tamaulipas (Universidad Autónoma de Tamaulipas). This study complied with the regulations and ethical considerations established for human research currently in effect in Mexico [32] and the international guidelines of the Helsinki Declaration [33].

### 2.5. Statistical Analysis

For the processing and analysis of the data, to obtain descriptive statistics, the Statistical Package for Social Sciences (SPSS version 25, IBM Corp, Armonk, NY, USA) was used. For the SEM, (AMOS version 25, IBM Corp, Armonk, NY, USA) was used. Asymmetry (±1.5) and kurtosis (±2.5) contrasts were obtained to confirm a multivariate normal distribution. To estimate the parameters, the maximum likelihood method was used. To assess the fit of the model, the chi-squared test (χ^2^, *p* > 0.05) and relative chi-squared test (χ^2^/df < 3) were used, and the goodness of fit was estimated using the root mean square error of approximation (RMSEA < 0.05), root mean square residual (RMSR = close to zero), and goodness-of-fit index (GFI > 0.95). The comparative goodness-of-fit index (CFI > 0.95) and the normed fit index (NFI > 0.95) were used as indicators of the comparative fit. Regarding the parsimony of the fit, the parsimonious normed fit index (PNFI = the greater the value, the greater the fit) and the parsimonious goodness-of-fit index (PGFI = the greater the value, the greater the parsimony) were calculated [34,35].

## 3. Results

Table 1 provides the sociodemographic characteristics of the sample. Most participants were female (94.5%) and married (80.3%), with a mean age of 55.4 years.

Almost half (44.3%) of the people with diabetes had low self-management, with 41.5% having some symptoms of depression, 35.5% having family dysfunction, and 77% not receiving social support. The average regular self-efficacy score for the sample was 7.09, and this is two points above the median score on the self-efficacy measure. Overall, the average exercise per week for the sample was slightly more than 1 h of exercise per week (µ = 66.97 min). The sample had an average of 6.45 years of formal education and an average of 10.34 years since being diagnosed with diabetes (Table 2).

For the SEM, the total score for each variable was used rather than its categorization, since the analysis requires variables with an interval level of measurement. Except for exercise, for which an asymmetry score of 1.97 and kurtosis of 5.74 were obtained, the other variables met the acceptable criteria (±1.5 and ±2.5), confirming a multivariate normal distribution.

Regarding the research objective, model 1 (replica of the hypothetical model) showed adequate fit indicators and was not significant (*p* = 0.358); however, the quadratic error index (RMSR = 28.359) was high (see Table 3) and thus could be improved. Given the above asymmetry and kurtosis values, the exercise variable was excluded and the analysis was repeated. As shown in Table 3, in the modified model (model 2), the GFIs were improved as follows: χ^2^/gl = 0.89 (*p* = 0.529), RMSEA = 0.000, and (CFI) = 1.000. These values indicate a very good fit and an acceptable degree of parsimony (PNFI = 409, PGFI = 317).

Figure 2 shows the effects of the variables that influence the self-management variable and their correlations. Depression, social support, self-efficacy, family functionality, years of formal education and years with a diagnosis had a combined effect that contributed to explaining the self-management of diabetes. The magnitude of the self-efficacy effect was the highest (2.93), and that of social support was the lowest (0.17). The size of the correlation between family functionality and social support was also remarkable (37.51). Notably, this model, with six predictor variables derived from the hypothetical model, was improved and obtained good fit indices (see Table 2), explaining 33.6% of the variance in self-management in patients living with diabetes.

## 4. Discussion

The research objective was to assess a hypothetical model to explain the self-management behavior of patients with type II diabetes through a SEM of the population of users of the State Health Department of Tamaulipas, Mexico. Corresponding to this objective, the results provided in Table 2 show that the original hypothetical model could be improved (model 1). Therefore, the following minimal modifications were applied to increase the fit. The exercise variable was excluded. As shown in Figure 2, the final model comprises the variables that were identified as predictors of self-management from the greatest to the smallest effect in the following order: self-efficacy, depression, years of formal education or schooling, family functionality, and years with a diagnosis and social support. Their correlations are also shown.

Based on previous studies, multiple concordances with the self-management predictors identified in the present study were detected. The international literature has identified family support and self-efficacy [7,8], years with the disease, education level [10] and depression [11,12]. In Mexico, as expected, previous studies have identified all the predictive variables detected in this study [18,19]. These studies agree that self-efficacy exerts the greatest influence on self-management.

Studies focusing on path models of variables that explain self-management in people with diabetes have identified a significant number of variables. However, only years with the disease (or diagnosis) and self-efficacy have been identified as predictors in previous international studies [7,15]. This finding can be attributed to the diversity of tools used in the different studies or differences in the cultural contexts in which these studies were performed. Advances in knowledge concerning Mexico and Latin America are scarce; therefore, the results of this research are of great importance.

Among the theoretical–empirical findings of greatest interest, the predictive effect of self-efficacy on self-management behavior was detected. This finding is consistent with the results of previous studies with different samples [18,19], thus confirming the theoretical–empirical relationship among these variables and indicating that self-efficacy is a cognitive element that precedes the self-management behavior related to the disease [19,21]. Complementing this finding, this relationship (self-efficacy–self-management) was influenced by interactions with the effects of depression, family functionality and years of formal education.

It is important to consider the negative effect of depression (−0.48), which was explained by the PHQ-8 score, with the following interpretation: lower levels of depression predict greater self-management in people with diabetes. This relationship has been documented and is consistent with previous studies [11,12,13]. Notably, in the present model, depression was found to be related to self-efficacy and family functionality.

Perceived social support from friends and family had a direct effect on self-management in the following direction: the greater the perception of social support, the greater the patient’s self-management. In addition, a close relationship with family functionality was documented, which also had a direct effect on self-management: higher family functionality scores were related to greater self-management of type II diabetes. This result is also explained by the individual and family self-management theory, in which it is proposed that the context influences the individual and their family during the self-management process in a positive way [6]. Notably, family functionality was also related to self-efficacy and depression. Finally, it was observed that the years of formal education and years with a diagnosis variables had direct effects on self-management behavior and were related.

Another result of interest was that the exercise variable was excluded from the predictive model. This result differs from other studies in which exercise was a predictor of self-management [18,19]. This discrepancy can be explained by the fact that in our preliminary studies, we worked with samples of patients with different noncommunicable diseases, such as diabetes, arterial hypertension and obesity, and not exclusively with patients with type II diabetes. This result can also be attributed to the low average exercise per week reported by the participants in this study (66 min per week), which could attenuate the effect of this variable on self-management in the people with diabetes who constituted the study sample.

The model clearly illustrates the direct and indirect effects of the independent variables on self-management. Clarity regarding such interactions can provide empirical support for the treatment of patients with type II diabetes or the design of programs that promote self-management and include topics related to family functionality, social support, depression and, in particular, self-efficacy, which has been shown to have an important effect on self-management behavior.

Finally, the results of this study contribute to addressing the serious problem of diabetes, which every society faces, in an attempt to mitigate its devastating consequences, epidemiological indicators and high cost to health systems [1,2].

One limitation of this study was that it involved a nonprobabilistic sample; therefore, the results cannot be extrapolated to the general population. It is also worth highlighting additional limitations, such as the small size of the sample in this study and the disproportionate number of women. Therefore, the results should be interpreted with caution. Similarly, it should be noted that the data collection was conducted by the researchers responsible for this study, which may have generated some bias in the results, an effect attributed to social instability [36]. Another limitation was the cross-sectional design, which prevents confirmation of the temporal stability of the model. Therefore, probabilistic samples should be utilized, and studies in this line of research should continue to be conducted to identify different variables that contribute to improving the understanding of self-management in relation to diabetes.

## 5. Conclusions

Based on the results obtained from the sample of patients with diabetes who participated in this study, the hypothetical model showed adequate fit indicators. However, abnormal distribution problems were detected in the exercise predictor variable. Once the exercise variable was eliminated, the improved model achieved a good fit, which allowed the explanation of self-management behavior through the following variables: self-efficacy, depression, years of formal education, family functionality, years with a diagnosis and social support. In addition, correlations were detected between the independent variables, allowing for clarification of their interactions. Therefore, in addition to the excellent goodness of fit, the improved model presented a good comparative fit and acceptable parsimony, representing an important advance in knowledge of self-management, particularly in the Latino population, which is rarely studied in terms of this topic.

The percentage of variance that is no longer explained in the model (66.4%) suggests the relevance of integrating other predictive variables that contribute to increasing these percentages and testing this model in different contexts, such as with patients with other diseases, patients from different ethnic groups or patients of different nationalities. Thus, it is suggested to use different strategies in the selection of the sample and the collection of data. In addition, this study provides theoretical–empirical evidence that supports the design of self-management programs to combat the health problems that diabetes presents.

The objective of studying diabetes self-management in a Latino population was met. In addition to self-efficacy being related to self-management, in this population the role of family functionality and social support (which were correlated) was significantly related to self-management. Depression and years of formal education were also related to self-management. The relationship of these variable to self-management suggests a role for improving family functionality and social support in Latino patients with diabetes. In addition, a focus on decreasing depressive symptoms is a worthwhile endeavor with this population. This study highlights how the combination of family social environment and individual mental health factors can be targeted to improve management of diabetes.

## Figures and Tables

**Figure 1 healthcare-11-01284-f001:**
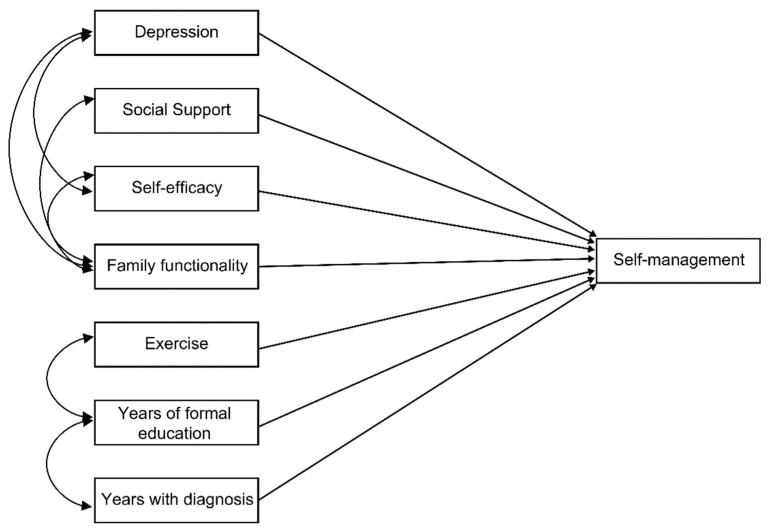
Hypothetical model of self-management behavior predictors in people with diabetes.

**Figure 2 healthcare-11-01284-f002:**
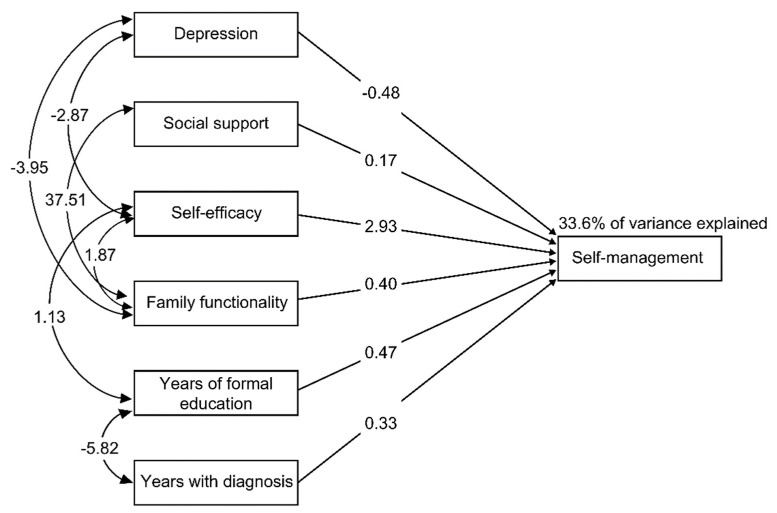
Final model of self-management behavior predictors in people with diabetes.

**Table 1 healthcare-11-01284-t001:** Sociodemographic characteristics of patients with diabetes.

n = 183	N	%	M	SD
Sex				
Male	10	5.5		
Female	173	94.5		
Marital status				
Married	147	80.3		
Widower	14	7.7		
Divorced	12	6.6		
Single	10	5.5		
Age			55.4	11.5

Note. n = absolute frequency, % = percentage, M = arithmetic mean, and SD = standard deviation.

**Table 2 healthcare-11-01284-t002:** Distribution of the variables that constitute the model.

n = 183	n	%	M	SD
Levels of self-management				
Low self-management	81	44.3		
Medium self-management	67	36.6		
High self-management	35	19.1		
Levels of depression				
No symptoms	107	58.5		
Mild symptoms	52	28.4		
Moderate symptoms	18	9.8		
Severe symptoms	3	1.6		
Serious symptoms	3	1.6		
Levels of family functionality				
Functional	118	64.5		
Mild dysfunction	29	15.8		
Moderate dysfunction	16	8.7		
Severe dysfunction	20	10.9		
Social support				
With social support	42	23.0		
No social support	141	77.0		
Self-efficacy			7.09	2.19
Exercise, in min.			66.97	85.55
Years of formal education			6.45	3.61
Years with a diagnosis			10.35	7.98
Total	183	100.0		

Note. n = absolute frequency, % = percentage, M = arithmetic mean, and SD = standard deviation.

**Table 3 healthcare-11-01284-t003:** Fit indicators of the self-management behavior predictive model in people with diabetes.

Model	Absolute Fit or Goodness of Fit						Comparative Fit		Parsimonious Fit	
	χ^2^	gl	χ^2^/gl	RMSEA	RMSR	GFI	CFI	NFI	PNFI	PGFI
1	16.364	15	1.09	0.022	28.359	0.977	0.992	0.917	0.491	0.407
2	8.054	9	0.89	0.000	4.251	0.987	1.00	0.954	0.409	0.317

Note. Indices: χ^2^ = likelihood ratio, chi-squared statistic; df = degrees of freedom; χ^2^/df = relative chi-squared; RMSEA = point estimation of the root mean square error of approximation; RMSR = root of the mean square residual; GFI = goodness-of-fit index; CFI = comparative fit index; NFI = normed fit index; PNFI = parsimony-normed fit index; and PGFI = parsimonious goodness-of-fit index.

## Data Availability

The datasets used during the current study are available from the corresponding author on reasonable request.

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
