# Peer review of "Psychosocial Factors Associated with Self-Management in Patients with Diabetes"

_healthcare, 2023, doi:10.3390/healthcare11091284_

Round 1

Reviewer 1 Report

Thank you for an interesting and professionally relevant article. I think it's well written, but could be even more with a few minor changes.

On page 3 you write about the hypothetical path model, designed based mainly on earlier evidence. The model is certainly plausible, but can you clarify how it was created. You mention that the variables were identified through linear regression analysis as independent predictors of self-management behavior in patients with noncommunicable diseases. But which noncommunicable diseases did these patients have? Please clarify this here as in the present study it was specifically about patients with diabetes. It is good that you come back to this in lines 290 and 291, but it needs to appear already here.

There are clear and good explanations for measuring instruments.

A weakness of the study is, as you yourself also point out in the disussuin section, the selection procedure. Think about the fact that you yourselves as authors of this study administered the instruments to patients with diabetes. Could it have affected the outcome? Describe how in the discussion section. Likewise, I want you to write in more detail in which way the outcome of the study was affected by not letting the respondents choose the place where they could answer the questionnaires themselves, without being supervised by the authors/researchers of this study.

Your results are valuable for the research area and future studies. It is interesting that this model, improved and obtained good fit indices explained 33.6% of the variance in self-management in patients living with diabetes. The reasons why it did not explain a larger percentage are probably several, one of which is the selection process. How do you view the fact that you excluded the exercise variable. I would like some reasoning on this in the discussion.

A convenience sample, only 183 respondents, a model that does not explain more than 33.6% of the variation is not broadly indicative of how psychosocial factors are associated with self-management in patients with diabetes, but can be cautiously transferred to a similar population.

A few small linguistic comments: On line 80, strike out the word and after the semicolon. On line 88, strike out the word the in the first part of the sentence "As the shown in the literature,...".

References: The articles that are in English seem to be very relevant to your script. If it is possible for you to have only references in English, it would increase the understanding of an international readership.

Author Response

Reviewer 1

Open Review

Quality of English Language

( ) English very difficult to understand/incomprehensible  
(x) Extensive editing of English language and style required  
( ) Moderate English changes required  
( ) English language and style are fine/minor spell check required  
( ) I am not qualified to assess the quality of English in this paper

Yes      Can be improved        Must be improved       Not applicable

Does the introduction provide sufficient background and include all relevant references?

(x)        ( )         ( )         ( )

Are all the cited references relevant to the research?

( )        (x)        ( )         ( )

Is the research design appropriate?

(x)        ( )         ( )         ( )

Are the methods adequately described?

( )        (x)        ( )         ( )

Are the results clearly presented?

(x)        ( )         ( )         ( )

Are the conclusions supported by the results?

(x)        ( )         ( )         ( )

Comments and Suggestions for Authors

Thank you for an interesting and professionally relevant article. I think it's well written, but could be even more with a few minor changes.

-On page 3 you write about the hypothetical path model, designed based mainly on earlier evidence. The model is certainly plausible, but can you clarify how it was created. You mention that the variables were identified through linear regression analysis as independent predictors of self-management behavior in patients with noncommunicable diseases. But which noncommunicable diseases did these patients have? Please clarify this here as in the present study it was specifically about patients with diabetes. It is good that you come back to this in lines 290 and 291, but it needs to appear already here.

Response:

Noncommunicable diseases were included, such as diabetes, arterial hypertension, and obesity.

Change:

In these studies, the following variables were identified through linear regression analysis as independent predictors of self-management behavior in patients with noncommunicable diseases (diabetes, arterial hypertension, obesity). Lines 91, 92

This discrepancy can be explained by the fact that in our preliminary studies, we worked with samples of patients with different noncommunicable diseases, such as diabetes, arterial hypertension and obesity, and not exclusively with patients with type II diabetes. Line 281

-There are clear and good explanations for measuring instruments.

-A weakness of the study is, as you yourself also point out in the disussuin section, the selection procedure. Think about the fact that you yourselves as authors of this study administered the instruments to patients with diabetes. Could it have affected the outcome? Describe how in the discussion section. Likewise, I want you to write in more detail in which way the outcome of the study was affected by not letting the respondents choose the place where they could answer the questionnaires themselves, without being supervised by the authors/researchers of this study.

Response:

In the last paragraph of the discussion, the application of instruments by the researchers of this work was addressed as another limitation that could generate bias in the results, which could be attributed to social desirability.

Change:

Similarly, it should be noted that the data collection was conducted by the researchers responsible for this study, which may have generated some bias in the results, an effect attributed to social instability [36]. Lines 297 - 300

Your results are valuable for the research area and future studies. It is interesting that this model, improved and obtained good fit indices explained 33.6% of the variance in self-management in patients living with diabetes. The reasons why it did not explain a larger percentage are probably several, one of which is the selection process. 

Response:

At the end of the conclusion, a suggestion associated with the percentage of variance explained by the model was integrated that addresses the selection of the sample and the data collection.

Change:

Thus, it is also suggested to use different strategies in the selection of the sample and the collection of data. Lines 321-323

How do you view the fact that you excluded the exercise variable. I would like some reasoning on this in the discussion.

Response:

In addition to the difference in the study sample from previous studies, an explanation has been included regarding the low average exercise per week reported by the study sample.

Change:

This result can also be attributed to the low average exercise per week reported by the participants in this study (66 min per week), which could attenuate the effect of this variable on self-management in the people with diabetes who constituted the study sample. Lines 280-284

A convenience sample, only 183 respondents, a model that does not explain more than 33.6% of the variation is not broadly indicative of how psychosocial factors are associated with self-management in patients with diabetes, but can be cautiously transferred to a similar population.

Response:

The above was taken into account in the discussion, and the conclusions were pointed out as a limitation in the study.

Change:

One limitation of the study was that it involved a nonprobabilistic sample; there-fore, the results cannot be extrapolated to the general population. It is also worth high-lighting additional limitations, such as the small size of the sample of this study and the disproportion by sex. Therefore, the results should be interpreted with caution. Lines 294-297

The percentage of variance that is no longer explained in the model (66.4%) sug-gests the relevance of integrating other predictive variables that contribute to increas-ing these percentages and testing this model in different contexts, such as with patients with other diseases, patients from different ethnic groups or patients of different na-tionalities. Lines 318-321.

A few small linguistic comments: On line 80, strike out the word and after the semicolon. On line 88, strike out the word the in the first part of the sentence "As the shown in the literature,...". 

Response:

The adjustments recommended by the reviewer were made.

Change:

education [10]; knowledge of the disease. Line 67

As shown in the literature. Line 75

References: The articles that are in English seem to be very relevant to your script. If it is possible for you to have only references in English, it would increase the understanding of an international readership.

Response:

Some of the references in the English language were modified; however, this was impossible for some articles published in Spanish.

Change:

Pan American Health Organization, World Health Organization. Innovative care for chronic condition, organizing and delivering high quality care for chronic noncommunicable diseases in the America. Available online: chrome-extension://efaidnbmnnnibpcajpcglclefindmkaj/https://www.paho.org/hq/dmdocuments/2013/PAHO-Innovate-Care-2013-Eng.pdf (accessed on 26 January 2023).

Pan American Health. Plan of Action for the Prevention and Control of Noncommunicable Diseases in the Americas 2013-2019.Available online: https://www.paho.org/hq/dmdocuments/2015/plan-accion-42prevencion-control-ent-americas.pdf. (accessed on 27 January 2023).

New References.

International Diabetes Federation. IDF Diabetes Atlas 10TH edición. Available online: (accessed on 23 March 2023). https://diabetesatlas.org/idfawp/resource-files/2021/07/IDF_Atlas_10th_Edition_2021.pdf

Ryan, P., Sawin, K.J. The individual and family self-management theory: background and perspectives on context, process, and outcomes. Nurs Outlook [Internet]. 2009 [Consultado 27 Marzo 2023]; 57(4): 2017-255. Available online:https://pubmed.ncbi.nlm.nih.gov/19631064/

Baptista DR, Wiens A, Pontarolo R, Regis L, Torelli WC, Januário CC. The chronic care model for type 2 diabetes: a systematic revie. Diabetol Metab Syndr [Internet]. 2016 [Consultado 27 Marzo 2023]; 8(7). Available online: https://pubmed.ncbi.nlm.nih.gov/26807158/

Vesely, S.; Klockner, C.A. Social desirability in environmental psychology research: three meta-analyses. Front Psychol. 2020, 11, 1395; DOI: 10.3389/fpsyg.2020.01395. eCollection 2020.

Submission Date

24 February 2023

Date of this review

07 Mar 2023 08:44:55

Reviewer 2 Report

The introduction section is way too long and needs to be shortened with a focus on the topic and aim of the paper.

Other than that, the study is novel and provides a good summary. Conclusions and discussion section uses too much statistical jargon that should be simplified for the readers who have no statistical background. Also, the English language can be improved with using less run-on sentences and use of better terms than "very good" for example. 

Author Response

Reviewer 2

Open Review

Quality of English Language

( ) English very difficult to understand/incomprehensible  
( ) Extensive editing of English language and style required  
(x) Moderate English changes required  
( ) English language and style are fine/minor spell check required  
( ) I am not qualified to assess the quality of English in this paper 

Yes      Can be improved        Must be improved       Not applicable

Does the introduction provide sufficient background and include all relevant references?

( )        ( )         (x)        ( )

Are all the cited references relevant to the research?

(x)        ( )         ( )         ( )

Is the research design appropriate?

( )        (x)        ( )         ( )

Are the methods adequately described?

( )        (x)        ( )         ( )

Are the results clearly presented?

( )        (x)        ( )         ( )

Are the conclusions supported by the results?

( )        (x)        ( )         ( )

Comments and Suggestions for Authors

The introduction section is way too long and needs to be shortened with a focus on the topic and aim of the paper.

Response:

The first 3 paragraphs were reduced to 1 paragraph with updated references. In addition, paragraph 6, cited with references 8 and 9, was deleted. With these changes, this section was reduced to 8 paragraphs.

Change:

First paragraph

Diabetes is a worrying health problem worldwide. It is estimated that in 2021, 537 million people lived with this disease, and an economic cost of 966 million dollars was generated [1]. The National Health and Nutrition Survey (2018-2019) revealed that 14.4% of older adults in Mexico have diabetes, and this percentage exceeds 30% in the population aged over 50 [2].Lines 44-48

Other than that, the study is novel and provides a good summary. Conclusions and discussion section uses too much statistical jargon that should be simplified for the readers who have no statistical background.

Response:

The term regression model was eliminated, and the term structural equation models was changed to path models. The authors consider the other terms necessary for the understanding of the manuscript

Change:

Based on previous studies, multiple concordances with the self-management predictors identified in the present study were detected. Line 242

Studies focusing on path models of variables that explain self-management in people with diabetes have identified a significant number of variables; however, only years with the disease (or diagnosis) and self-efficacy have been identified as predictors in previous international studies [7,15]. Line 248

Also, the English language can be improved with using less run-on sentences and use of better terms than "very good" for example. 

Response:

Changes were made based on the reviewer's suggestion.

Change:

The goodness-of-fit indices (GFIs) were good, Linea 33

good internal consistency. Line 134

good internal awareness was obtained. Líne 145

the improved model achieved good fit. Line 309

Therefore, in addition to the excellent goodness of fit, the improved model presented good comparative fit and acceptable parsimony (Lines 313 and 314).

Reviewer 3 Report

This manuscript is an interesting contribution to explanatory models; however, there are some inaccuracies listed below:

1) Lines 60-64: The authors refer to a model of chronic care that comprises 6 components, but no explicit reference is made to the Chronic Care Model (Does it infer from the text that they refer to this model? If not, the authors should clarify which model they refer to. Also, I find it helpful to include references to the Chronic Care Model).

2) Intro: No reference is made to Individual and Family Self-Management Theory (IFSMT) (Ryan, P., & Sawin, K. J. 2009. The Individual and Family Self-Management Theory: background and perspectives on context, process, and outcomes. Nursing outlook, 57(4), 217–225.e6. https://doi.org/10.1016/j.outlook.2008.10.004); please implement.

3) Materials and Methods - 2.1. Participants: The authors must justify the calculation of the sample size and, if no calculation has been made, they must declare and justify it.

4) Table 2 (layout): It is necessary to separate the categories with horizontal lines; the table is visually chaotic.

I remain at your disposal for further clarifications.

Best regards.

Author Response

Reviewer 3

Open Review

Quality of English Language

( ) English very difficult to understand/incomprehensible  
( ) Extensive editing of English language and style required  
( ) Moderate English changes required  
(x) English language and style are fine/minor spell check required  
( ) I am not qualified to assess the quality of English in this paper 

Yes      Can be improved        Must be improved       Not applicable

Does the introduction provide sufficient background and include all relevant references?

( )        (x)        ( )         ( )

Are all the cited references relevant to the research?

(x)        ( )         ( )         ( )

Is the research design appropriate?

(x)        ( )         ( )         ( )

Are the methods adequately described?

( )        (x)        ( )         ( )

Are the results clearly presented?

(x)        ( )         ( )         ( )

Are the conclusions supported by the results?

(x)        ( )         ( )         ( )

Comments and Suggestions for Authors

This manuscript is an interesting contribution to explanatory models; however, there are some inaccuracies listed below:

  • Lines 60-64: The authors refer to a model of chronic care that comprises 6 components, but no explicit reference is made to the Chronic Care Model (Does it infer from the text that they refer to this model? If not, the authors should clarify which model they refer to. Also, I find it helpful to include references to the Chronic Care Model).

Response:

The name of the chronic care model was changed, and a reference to the model applied in diabetes was included.

Change

Faced with the problems that this disease presents, the Pan American Health Organization promotes the Chronic Care Model in the Americas that comprises 6 components, including support for self-management, which is conceptualized as patient education and health interventions by personnel and can strengthen skills for the proper management of this chronic disease [3, 4]. Line 50

  • Intro: No reference is made to Individual and Family Self-Management Theory (IFSMT) (Ryan, P., & Sawin, K. J. 2009. The Individual and Family Self-Management Theory: background and perspectives on context, process, and outcomes. Nursing outlook, 57(4), 217–225.e6. https://doi.org/10.1016/j.outlook.2008.10.004); please implement.

Response

The article suggested by the reviewer was included.

Change

In addition, it is important to consider the individual and family processes that occur in the context and that can act as risk or protective factors for the development of good self-management [6]. Lines 60-62

  • Materials and Methods - 2.1. Participants: The authors must justify the calculation of the sample size and, if no calculation has been made, they must declare and justify it.

Response

Added signage responding to author and in discussion referenced to implications of sample size and sampling

Change

All patients who agreed to participate in the study in the period from November 2018 to March 2019 were recruited; therefore, the sample size was not calculated. Lines 107-109

One limitation of the study was that it involved a nonprobabilistic sample; therefore, the results cannot be extrapolated to the general population. Lines 294-295

  • Table 2 (layout): It is necessary to separate the categories with horizontal lines; the table is visually chaotic.

Response

The change was made.

Change

n = 183

 n

 %

 M

 SD

 Levels of self-management

 Low self-management

81

44.3

 Medium self-management

67

36.6

 High self-management

35

19.1

 Levels of depression

 No symptoms

107

58.5

 Mild symptoms

52

28.4

 Moderate symptoms

18

9.8

 Severe symptoms

3

1.6

 Serious symptoms

3

1.6

 Levels of family functionality

 Functional

118

64.5

 Mild dysfunction

29

15.8

 Moderate dysfunction

16

8.7

 Severe dysfunction

20

10.9

 Social support

 With social support

42

23.0

 No social support

141

77.0

 Self-efficacy

7.09

2.19

 Exercise, in min.

66.97

85.55

 Years of study

6.45

3.61

 Years with diagnosis

10.35

7.98

 Total

183

100.0

I remain at your disposal for further clarifications.

Best regards.